# MessageNet: Message Classification using Natural Language Processing and Meta-data

## Abstract

In this paper we propose a new Deep Learning (DL) approach for message classification. Our method is based on the state-of-the-art Natural Language Processing (NLP) building blocks, combined with a novel technique for infusing the meta-data input that is typically available in messages such as the sender information, timestamps, attached image, audio, affiliations, and more. As we demonstrate throughout the paper, going beyond the mere text by leveraging all available channels in the message, could yield an improved representation and higher classification accuracy. To achieve message representation, each type of input is processed in a dedicated block in the neural network architecture that is suitable for the data type. Such an implementation enables training all blocks together simultaneously, and forming cross channels features in the network. We show in the Experiments Section that in some cases, message's meta-data holds an additional information that cannot be extracted just from the text, and when using this information we achieve better performance. Furthermore, we demonstrate that our multi-modality block approach outperforms other approaches for injecting the meta data to the the text classifier.

## 1 Introduction

Many real world applications require message classification and regression, such as handling spam emails Karim et al. (2020), ticket routing Han et al. (2020), article sentiment review Medhat et al. (2014) and more. Accurate message classification could improve critical scenarios such as in call centers (routing tickets based on topic) Han et al. (2020), alert systems (flagging highly important alert messages) Gupta et al. (2012), and categorizing incoming messages (automatically unclutter emails) Karim et al. (2020); Klimt & Yang (2004). The main distinction between text and message classification is the availability of additional attributes, such as the sender information, timestamps, attached image, audio, affiliations, and more. New message classification contests often appear in the prominent platforms (i.e., Kaggle Kaggle), showing how this topic is sought after. There are already many data-sets to explore in this field, but no clear winner algorithm that fits all scenarios with high accuracy, efficiency and simplicity (in terms of implementation and interpretation).

A notable advancement in the field of NLP is the attention based transformers architecture Vaswani et al. (2017). This family of methods excels in finding local connections between words, and better understanding the meaning of a sentence. A leading example is the Bidirectional Encoder Representations from Transformers (BERT) Devlin et al. (2018) as well as its variations Liu et al. (2019); Lan et al. (2019); Sanh et al. (2019), winning certain benchmarks Rajpurkar et al. (2018); Wang et al. (2019). Several packages, such as Huggingface Transformers Wolf et al. (2019), make such models accessible and easy to use as well as provide pre-trained versions. In addition, one can use transfer learning Pan & Yang (2009) to further train BERT on their on data, creating a tailored model for the specific task at hand.

BERT, and often other transformer based models, are designed to handle text. They operate on the words of a given text by encoding them into tokens, and by the connections between the tokens they learn the context of sentences. This approach is limited, since sometimes more information can be extracted and used, not necessarily textual. Throughout this paper we refer to this information as meta-data to distinguish it from the main stream of textual content (though one may recognize it as the core data, depending on the application). For example, a meta-data could be the time stamp

of when the text was written, sent, published, etc. Another example is the writer of the text, when dealing with a small list of writers of a corpus. There have been some attempts to incorporate these into BERT models, for example by assigning artificial tokens for writers or for temporal segments (token per month for example) Zhang et al. (2021). This approach is limited since not all meta-data entries are suitable for encoding by tokenization. In the example of temporal segments, more segments introduce more tokens, leading to large computational resources consumption, and less segments cause loss of information. Another approach is to concatenate the embeddings, created by the transformer module, with the outputs of an embedding module for the meta-data. In this approach, a transformer for the text is trained (using direct or transfer learning) on the text, and other separate modules (time series embedding, senders embeddings, etc.) are used to embed the meta-data. All the embeddings are then concatenated and used as inputs to a classification network. A drawback of this approach is that the internal network features are not trained from a combination of diffident input streams, and therefore avoid cross dependent features (e.g. the importance of an email is not only determined by its content, but also by who sent it, when, to whom else, attachments, etc.).

To bridge these gaps, we implement a transformer based model that is able to train with both the text (transformer architecture) and meta-data. We create a new architecture of a blocks based network. Each block handles different kind of inputs. Splitting to blocks enables the flexibility to handle different kind of inputs. We present results of the method with a main block based on a transformer that handles the text, and an additional block that handles the pre-processed meta-data inputs individually. This method can be extended to support more complex blocks, such as an advanced DL model for images Wang et al. (2017), a temporal analysis block to extract information from temporal meta-data Ienco & Interdonato (2020), additional transformer blocks for multiple text inputs (for example, subject and body of an email), categorical data, and more. To demonstrate the performance of the method we run multiple experiments on publicly available data-sets to show the advantages of using the block architecture, and compare them to the transformer benchmark (BERT), Random Forest (RF) classifier, and Multi-Layer Perceptron (MLP) networks. We achieve competitive results, and in most cases lead those benchmarks, showcasing that there is much to extract from the meta-data compared to just using text for classification tasks.

## 2 RELATED WORK

**Natural language processing tasks.** The publication of BERT Devlin et al. (2018) has been a turning point in the text classification domain. The authors demonstrated high accuracy on complicated tasks such as question and answer, named entity recognition, and textual entailment Wang et al. (2019). Since then, many authors investigated improved architectures and variations such as RoBERTa Liu et al. (2019), ALBERT Lan et al. (2019), DistilBERT Sanh et al. (2019), and more. Some focus on better performance on the benchmark tasks, and some create lighter versions of the model that reduce the computational demands while preserving competitive accuracy. Other propositions, like XLNet Yang et al. (2019) and GPT-3 Brown et al. (2020), introduce competing architectures to BERT (also using transformers). The benchmarks for these models are commonly GLUE, SuperGLUE Wang et al. (2019), SQuAD 2.0 Rajpurkar et al. (2018), and more. Text classification is a less common benchmark, but the models can be used for this task as shown in this paper.

**Accessibility of transformers.** Another contributing factor to the growing popularity of transformers is the variety of open-source code bases that make it easy for data-scientist to experiment with different architectures and then use it in their applications. The Huggingface transformers package Wolf et al. (2020) is a Python library that can be used to train and fine-tune models, with a large variety of base models to choose from, and straightforward implementation. The GPT-3 Brown et al. (2020) has been published as open source and, similar to several other implementations, offers a convenient application programming interface (API). We mention that many libraries that do not use machine-learning for text classification exist such as NLTK Bird et al. (2009), spaCy Honnibal & Montani (2017), and more. These are also easily accessible and offer advanced NLP feature extraction and other text analysis tools.

**Text classification.** There are many tasks in text classification, and each may be considered as a field of study. A popular one is sentiment analysis, aiming to classify texts as positive or negative.

The survey in Medhat et al. (2014) presents the challenges in this domain and the latest innovations. Another example is the Spam or Ham task, where one tries to differentiate a relevant email from irrelevant ones (like advertisements, phishing attempts, etc.) Karim et al. (2020). In this work we investigate multi-label classification of messages. For example, classifying the category of a product based on purchase review, the category of a thread based on the posts, the culinary specialty of a restaurant from customer reviews, the type of product from purchase feedback, the category of an incoming email, and so on. For each of these tasks, publicly available data-sets exist and are used in this work to quantify the success of the proposed method. In addition, there are many competitions in Kaggle Kaggle and other machine-learning research benchmark websites, using these data-sets.

**Message classification with meta-data.** There are two commonly used method to incorporate meta-data with textual information for message categorization when using transformers. The first is concatenating the embeddings, computed by the transformer, with the outputs of other embedding systems that are built specifically for the meta-data. In Rahman et al. (2021), the authors use this approach for visual and audio meta-data. In Ostendorff et al. (2019) the authors use properties of the text as meta-data and show that this approach can also work for the German language. In Xu et al. (2020) the meta-data is the layout information of scanned documents, and the authors propose an innovative architecture to extract information from both text and layout information. There are many other studies exploring this approach. While it is simple to implement, this strategy has several drawbacks. The training is usually done independently for the text and the meta-data, and the decisions are made as a "voting between classifiers" approach. This may lead to conflicts, since the response of the text to the label may be different from the response of the meta-data and the label, resulting in very low confidence predictions. We compare the performance of the proposed method to this one in the results section. The second popular approach is to assign tokens to the meta-data, and add this to the tokenized input of the transformer. In Zhang et al. (2021), in addition to a hierarchy of labels, the authors introduce a method to inject multiple meta-data inputs with varying types (web, references, etc.) as tokens to the embedding vector. Due to the simplicity of embedding the information using tokens, in terms of algorithm and implementation, developers use this approach in their codes and it appears in many online notebooks and blogs (open source codes) as well. The main drawback of these methods are the robustness to the input data. For example, representing an image as a series of tokens is either done by encoding the image which usually faces loss of information, or by utilizing a large number of tokens that exponentially highers the computational cost. In the numerical experiments presented here, we do not compare to this method since the feature extraction we are using has a varying and potentially high number of features, which lead to computational resource exhaustion when using this method. In this work, we propose a method that can address the issues of the two popular methods, as described in the next section.

## 3 APPROACH

We propose a method based on blocks to train a linguistic model with meta-data for a specific text classification task. By splitting each type of meta-data input into different blocks, one can use state-of-the-art deep-learning architectures to handle each meta-data type uniquely and more efficiently. In addition, the training is done using all block and in a unified training loop, adjusting all the weights of all blocks in every optimizer step, so all information from the text and meta-data sources contributes to the learning process.

### 3.1 BLOCKS ARCHITECTURE

The transformer models, including BERT, can be used for text classification with the input text and corresponding output labels. However, we claim that a lot of information can be found in the meta-data of the text. as can be seen in Figure 1, we use the transformer model as a single block of a neural network. Then, we can add additional blocks for dealing with the meta-data inputs.

With the recent developments in deep-learning, there are many advanced method of extracting information from input signals for classification. For example, in Ang et al. (2020) the authors discuss ways to use deep-learning for analysing time series data. Messages typically have a temporal element, such as the time of arrival of an email, the time when a review has been posted, a paper has been published, etc. We propose to utilize these advancements together for better model training.

In Figure 1, an overall schematic view of the proposed approach is presented. In the first row, a standard transformer architecture is illustrated Wolf et al. (2020). The inputs are the tokenized messages, followed by the transformer layers that are initially pre-trained. These layers are further trained (using transfer learning) to produce the embeddings, and a classification layer is used to predict the categories of the messages. The transformer layers in this illustration are presented using dashed lines to express the transfer learning process. The second row presents a meta-data extractor using a deep-neural network. The green layers express layers trained for classification (for example, fully-connected network architecture). The third row presents another transfer learning architecture. Each row expresses a different block of a neural-network that handles different meta-data inputs.

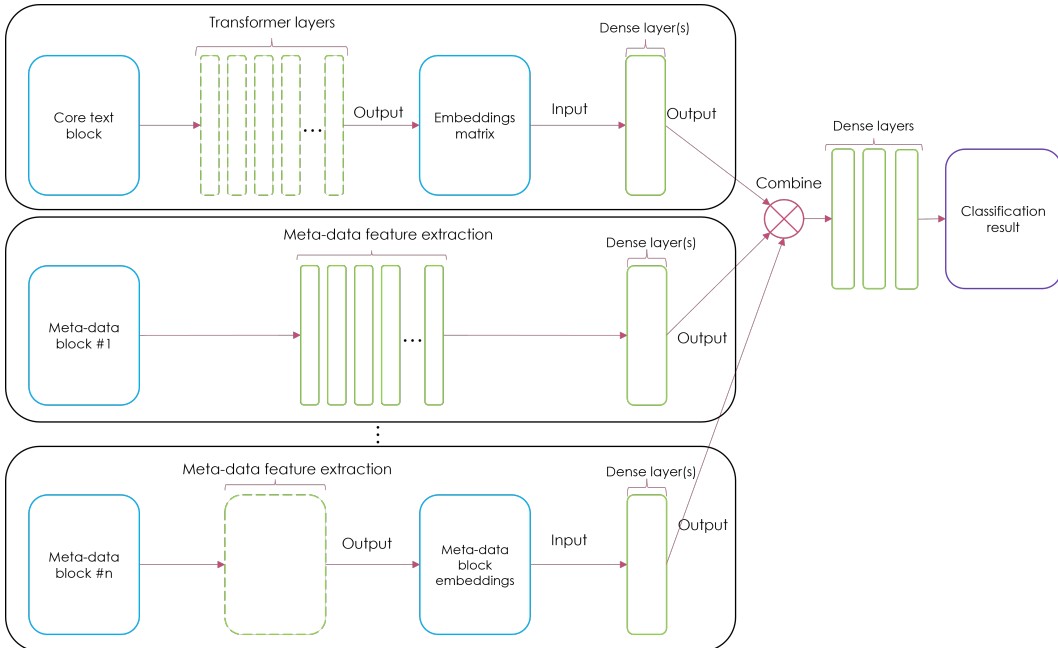

Figure 1: A sketch diagram of the block method. The first row illustrates a transformer architecture to handle textual input, while the second and third rows illustrate neural networks for handling a specific meta-data input. All the modules are then combines to create a unified prediction. Blue tiles are static and green tiles are trainable layers. The dashed green tiles illustrate the transfer learning layers

### 3.1.1 REPRESENTATION

Using the propose approach, the expected representation of the data is much different than the one achieved by the prominent approaches mentioned in section 2. Since the blocks are trained simultaneously, information from the meta-data may impact the training of the core textual block (the transformer block), and vice-versa. Therefore, a new and different representation is achieved. As a toy illustrative example, let us say that a single sample has text suggesting a specific class but meta-data suggesting another, the textual block would train differently (and the textual representation would be different, taking this in account), and the weights of the textual block would be able to capture this difference. We illustrate the different in Figure 2. In Figure 2a we illustrate using tokens to inject the meta-data into the transformer layers, producing an embedding (e1-eM1). In Figure2b we demonstrate concatenation of the embedding produced by the transformer layers (e1-eM2) with a vector representation (v) of the meta-data. In Figure 2c the proposed blocks approach is illustrated. The trainable blocks are wrapped in green to emphasize that training is done together, creating a unified representation of the message, compared to 2b where the transformer layers and the dense layers are trained independently, and the output representations are concatenated. We emphasize that the blocks may have different architectures and may produce different vector representations, to produce a better representation.

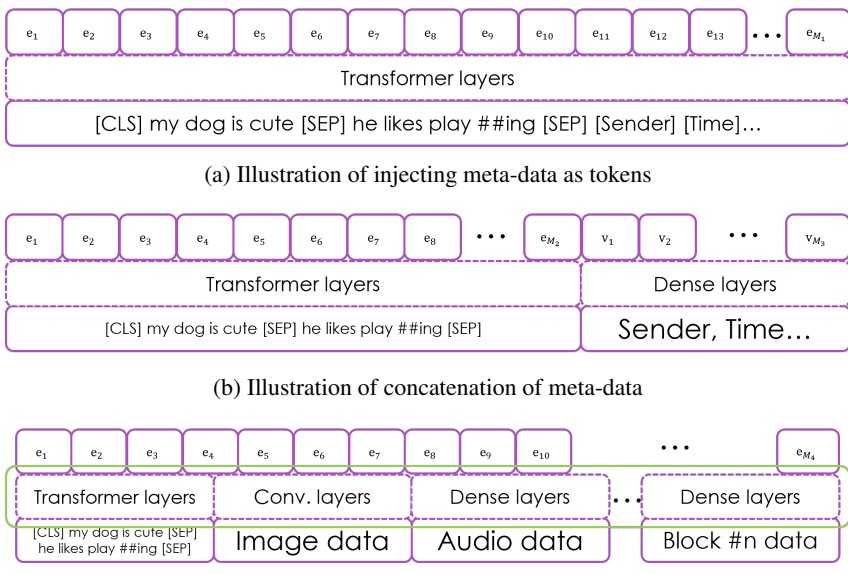

(a) Illustration of injecting meta-data as tokens

(b) Illustration of concatenation of meta-data

(c) Illustration of the blocks approach

Figure 2: Overview of the two common methods (a, b) and the proposed approach (c), emphasizing that the blocks are trained together, so information from one block may affect the training of the other blocks (as opposed to (b) for example, where the blocks are trained separately and concatenated)

### 3.1.2 COMBINE

After the blocks, we propose to take a combination of the layers (e.g. averaging the outputs, summing the outputs, concatenating the outputs, etc.). This combination may also be trained as part of the training loop. From the experimental tests we find that the best performance was achieved by having the combine step as a weighted concatenation of the outputs, producing an output sized as the number of classes times the number of blocks. We then use a few dense layers to produce an output sized as the number of classes, and apply softmax activation for the classification.

### 3.2 TRANSFER LEARNING

The text classification blocks often use transformer models. These models have complex architectures that achieve high performance on the available benchmark data-sets. To utilize the strength of these models, one may train a network with the same architecture but with their specific data-set. However, this tends to have high computational costs, and require large volume of data which is not so often available. Therefore, many of the state-of-the-art transformer models, including BERT, have an open source pre-trained versions available online, where one can download the pre-trained version and use a small data-set to train the model for only a few more iterations on that data. Since the pre-trained model already trained for a long time to understand textual embeddings, this transfer learning method adapts the model to the new data-set with lighter training (relative to pre-training).

Transfer learning is applicable for image data-sets, audio data-sets, time series analysis, and many more Pan & Yang (2009). The core block in the proposed method is operating on text, and the current stat-of-the-art methods for text classification are transfer learning, usually with transformers. That said, one may choose to implement other blocks using transfer learning as well. Using the proposed approach, the transfer learning can be done on all blocks simultaneously.

## 4 EXPERIMENTS

For the numerical experiments we train the network of the proposed approach with two blocks: a) the transformer block operating on the main text, and b) the meta-data block which is a fully-connected block operating on a one dimensional meta-data vector. We emphasize that more advanced blocks

can be used, but even this simple architecture provided results that demonstrated the value from the meta-data and the blocks approach. In addition, it is simple to distinguish between the transformer only architecture, and the transformer and meta-data architecture in this way, giving us better explainability of the contribution of the meta-data.

## 4.1 DATA-SETS

**Amazon reviews.** The Amazon product reviews data-set He & McAuley (2016) contains a large number of product reviews, varying by product category (label). The main text is the description of the review. We extract meta-data from the Amazon data-set comes from the reviewer field (sender), the review creation date and time (timestamp), and the overall satisfaction (enumerated). This data-set has in total 82.83 million product reviews. We sub-sample a subset of the reviews, by choosing the first 100,000 reviews of each category and then selecting the ones with the longest texts. We eventually save 4,687 reviews for training, 521 for validation, and 2,064 for testing (eliminating text-less messages). The final data-set is close to balanced (roughly the same number of reviews per category).

**Yelp Open Data-set.** The Yelp restaurant reviews data-set Yelp (2014) contains restaurant reviews, varying by culinary class (label). The main text is the restaurant review. We extract meta-data from the reviewer field (sender), the review creation date and time (timestamp), is the review useful/funny/cool (enumerated), and the number of stars (enumerated). We clean the data-set similar to the Amazon one, and save 4,612 reviews for training, 513 for validation, and 2,050 for testing, roughly balanced as well.

**Reddit.** The Reddit data-set Baumgartner et al. (2020) contains a large volume of Reddit posts, each belongs to a specific sub-Reddit (label). Specifically, we use the data-set of version 2 from 2010. The main text is the post content. We extract meta-data from the post uploader field (sender), the post upload date and time (timestamp), whether the post has comments (enumerated), and whether it has attachments (enumerated). We clean the data-set as well and save 4,950 posts for training, 550 for validation, and 2,200 for testing. We focus on 12 sub-Reddits (12 unique labels) and the data-set is roughly balanced.

**Enron Email Data-set.** The Enron email data-set Klimt & Yang (2004) contains a corpus of emails. Although this data-set is publicly available, we had access to a limited internal version that has been tagged based on email category (label). The main text is the email body. We extract meta-data from the email sender field (sender), and the email reception date and time (timestamp). We save 4,500 emails for training, 500 for validation, and 2,000 for testing. This data-set is not balanced.

## 4.2 FEATURE EXTRACTOR

To conduct the numerical experiments we first discuss a genuine feature extractor we use to extract the meta-data information from the available fields of the data-set. For each type of meta-data we extract corresponding features. Since different data-sets have different fields, we use generic modules to extract information from different fields of the same purpose, such as sender field (in emails data-set) or reviewer field (in restaurant reviews data-set). An overview of the meta-data features we extract from the data-set is given in Table 1. Note, that the different data-sets have different fields. Some of them do not have an enumerated field, and some have more than one.

## 4.3 RESULTS

To demonstrate the success of the proposed framework, we use a simple 2-blocks network. The first block is a transformer based block, specifically a pre-trained BERT model, and 2 fully-connected layers. That way we can examine the contribution of the meta-data carefully. We emphasize that with more advanced handling of the meta-data and introduction of more blocks, higher accuracy is expected. The transformer models are implemented using the Huggingface Transformers package Wolf et al. (2019). In addition, we used the ktrain package, specifically the text classification modules, and re-wrote them to have a message classification class.

We use the data-sets mentioned in Section 4.1. We compare multiple methods in terms of accuracy:

1. Pre-trained BERT with an additional fully-connected layer.

| Field type | Features | Number of slots | Description |
|---|---|---|---|
| Sender | Top senders | 120 | One hot vector, each slot represents if the sender is one of the 100 to senders or not |
| | Top affiliations | 120 | Same with sender top affiliations |
| | Sender frequency | 1 | Compute the frequency of messages received from this sender |
| Timestamp | Day | 7 | One hot vector, each slot represents if the message arrived at that day |
| | Working hours | 1 | Indicating if the message arrived within working hours or not |
| | Rush hour | 50 | Creating a histogram of 50 bins and a corresponding 50 slots one hot vector where each slot represents if the message arrived in the corresponding bin time |
| Enumerated | Enumerated value | #Options | One hot vector with 1 in the slot representing the option and 0 otherwise |
| Numeric | Numeric value | 1 | The numeric value |

Table 1: Review of meta-data features extraction per field

2. Pre-trained BERT with a random forest classifier.

3. Pre-trained BERT with concatenated meta-data and an additional fully-connected layer.

4. Pre-trained BERT with concatenated meta-data and a random forest classifier.

5. Transfer learning BERT with an additional fully-connected layer.

6. Transfer learning BERT with a random forest classifier.

7. Transfer learning BERT with concatenated meta-data and an additional fully-connected layer.

8. Transfer learning BERT with concatenated meta-data and a random forest classifier.

9. The proposed method (BERT and meta-data) with output averaging.

10. The proposed method (BERT and meta-data) with output weighting.

Methods 1-4 use embeddings from a pre-trained BERT model (without transfer learning), and train either an extra fully-connected layer or a random forest classifier Breiman (2001) for the classification. We do this either without (1-2) or with (3-4) concatenation of the meta-data to the embeddings, following the common methods in the literature. Methods 5-8 are similar, but the BERT transformers layers are further trained on the specific task data. Methods 3,4,7,8 are the reference methods of concatenating the BERT model embedding with a meta-data embedding. Methods 9-10 are the proposed ones, exploring the effect of combining the transformer output with the meta-data block output, once with averaging the block outputs and once with a fully-connected layer to learn the weights after concatenating the output representations of each block. The latter involves training to learn this weight, which is implemented so that it happens in the same training process as all other weights in the network. The results are given in Table 2. By observing the results, we see that the proposed method is competitive with all other methods and even outperforms most of them.

An interesting result is observed for the Enron emails data-set. We notice that the pre-trained methods performed better than the transfer-learning based methods. This can be explained by the mismatches between the textual information and the meta-data information. While the text suggests one class, the meta-data may suggest a different one. Methods 1-8, in this case, act as voting between classifiers. However, as mentioned in section 3.1.1, the proposed method is influenced by both text and meta-data and learns a better representation that can take into account these differences, performing better than all other methods.

| Method# | Amazon reviews | Yelp Open Data-set | Reddit | Enron Emails |
|---|---|---|---|---|
| 1 | 0.66 | 0.29 | 0.56 | 0.49 |
| 2 | 0.61 | 0.22 | 0.52 | 0.49 |
| 3 | 0.65 | 0.24 | 0.46 | 0.48 |
| 4 | 0.61 | 0.22 | 0.5 | 0.5 |
| 5 | 0.74 | 0.39 | 0.61 | 0.47 |
| 6 | 0.73 | 0.38 | 0.6 | 0.47 |
| 7 | 0.71 | 0.38 | **0.62** | 0.47 |
| 8 | 0.73 | 0.39 | 0.6 | 0.47 |
| 9 | 0.7 | 0.3 | **0.62** | 0.47 |
| 10 | **0.77** | **0.4** | **0.62** | **0.53** |

Table 2: Results table

## 5  CONCLUSION

In this paper we proposed a new framework for training classification models. The proposition relies on the availability of additional, not necessarily textual, data channels such as attached images, audio, sender and timestamp, etc. We proposed an architecture with which one can utilize the aforementioned additional information using different blocks, along with the text, and train a neural network to perform message classification more accurately. We demonstrate the strength of this method using a set examples, varying by data-set and classification algorithm, and show that the proposed method outperforms the reference related works.

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
