# OpenReview forum: "MESSAGENET: MESSAGE CLASSIFICATION USING NATURAL LANGUAGE PROCESSING AND META-DATA"
_ICLR.cc/2023/Conference — Submitted to ICLR 2023_

### Official Review · Reviewer_rydk · 2022-10-23

**Confidence:** 5
**Correctness:** 2
**Technical Novelty And Significance:** 1
**Empirical Novelty And Significance:** 1
**Recommendation:** 1

**Clarity, Quality, Novelty And Reproducibility:**

The paper is not clear, and the work does not rely on what has been done before on these popular benchmark datasets. Many details lack for reproducibility (hyperparameters specifically). The work is not novel enough.

**Strength And Weaknesses:**

Strengths:
- Good problem to tackle and many datasets.

Weaknesses:
- There is no need to reserve a subsection (3.2) or a paragraph in Related Work (Accessibility of Transformers) to explain the use of pre-trained models, as this is of widespread use. Please reserve the technical details for a "Training Details" section under Experiments, and remove them from everywhere else for easier reading.
- There should be a discussion of the results: are we using macro-F1 score or accuracy? Some of the results are very close together, and there should be an analysis to show why. For example, which meta-data features are important and which ones are not?
- There is no comparison with prior work, but rather a table of results with small technical changes.

Additional notes:
- Please fix the typos, as there are many repeated words and spelling mistakes
- Please fix the citations -- most of them should be in parentheses (those that are included as part of the discourse should not)

**Summary Of The Paper:**

This paper proposes to use meta-data for message classification. Specifically, the authors extract meta-data from popular benchmarks, and add them to the models used in training. The results show improvement, but the reader is not sure of the metrics used. The method resembles multi-modal models that already exist, and does not compare to existing methods (e.g. other papers).

**Summary Of The Review:**

The paper proposes a simple multi-modal method to add meta-data information to message classification. The authors do not compare to existing data, and there is no analysis or discussion of the results. The method is not novel, and the paper overall could use some proofreading.

---

### Official Review · Reviewer_7oAd · 2022-10-24

**Confidence:** 4
**Correctness:** 3
**Technical Novelty And Significance:** 1
**Empirical Novelty And Significance:** 1
**Recommendation:** 3

**Clarity, Quality, Novelty And Reproducibility:**

This paper is relatively clear but lacks novelty. Based on Sec. 3 and Sec. 4, it will be difficult to reproduce their results.

**Strength And Weaknesses:**

Strengths:

* This paper leverages additional information from the metadata to boost their model performance.
* Their multi-modality building block outperforms other methods.

Weaknesses:

* Leveraging metadata and ensembling BERTs have been used for both research prototyping and industry best practices. I am not sure where the novelty is.
* Using additional information from the metadata can introduce biases. I don't see any effort from the authors to debias their model and data.

**Summary Of The Paper:**

This paper proposes a new approach to message classification. Specifically:

* It is based on SOTA NLP building blocks.
* It has a novel technique to infuse metadata.

This paper shows that:

* Adding metadata increase the performance of their model.
* Their ulti-modality building block outperforms other methods.

**Summary Of The Review:**

This paper is clear but lacks novelty. I am leaning towards rejecting it.

---

### Official Review · Reviewer_uH2i · 2022-10-24

**Confidence:** 5
**Clarity, Quality, Novelty And Reproducibility:** The work is fairly straightforward an…
**Correctness:** 2
**Technical Novelty And Significance:** 1
**Empirical Novelty And Significance:** 1
**Recommendation:** 1

**Strength And Weaknesses:**

In my opinion, this paper is not at level of an ICLR paper. The ideas are fairly straightforward and probably among the first things attempted by ML practitioners at any single task. I do not think there are any significant technical ideas one can take from this paper.

**Summary Of The Paper:**

The paper introduces a mechanism to combine text and metadata in messages. The authors propose using a "block" (a custom architecture or pipeline) for each metadata field which is supposedly superior to placing metadata as [SEP] separated sentences or a single pipeline for the whole metadata.

**Summary Of The Review:**

- Paper presents a very trivial architecture as an "improvement" over various BERT based pipelines. The architecture is really not that sophisticated and seems like a straightforward thing any practitioner would try. No real technical insights.

---

### Official Review · Reviewer_FKbD · 2022-10-24

**Confidence:** 4
**Correctness:** 2
**Technical Novelty And Significance:** 1
**Empirical Novelty And Significance:** 1
**Recommendation:** 3

**Clarity, Quality, Novelty And Reproducibility:**

This submission in its current form might need a bit of technical guidance in terms of content organization and experimental design.
For now readers can only see an architecture being presented and empirical results were shown from small variants (e.g., the choice of final head to be a random forest or standard MLP) of the presented framework.

The presented framework is standard BERT with augmented input, which does not seem to be technically novel.

**Strength And Weaknesses:**

Strength:
- The use of metadata makes sense in some scenarios

Weaknesses:
- The motivation does not seem to be clear on the specific benchmarks: why would the time or sender information useful for classifying product categories or topic or other categorical labels?
- Readers might not learn much information from this submission since the presented framework is rather standard while one can draw very few conclusions from the current experiments
- No comparison with task-specific baselines: the specific datasets have all been used in previous studies while there are no comparisons on previous results, making the last sentence of the conclusion section poorly justified.


**Summary Of The Paper:**

This paper presents a framework for message classification, built with standard transformer (or more precisely, BERT, as used in all experiments) along with meta-data as augmented input. Experiments were conducted on standard text datasets with meta-data of sender and timing information.

**Summary Of The Review:**

Not too much technical novelty to be presented at a top ML/NLP conference

---

### Decision · Program_Chairs · 2023-01-20

**Decision:**

Reject

**Justification For Why Not Higher Score:**

The proposed method is not novel.

**Justification For Why Not Lower Score:**

N/A

**Metareview: Summary, Strengths And Weaknesses:**

The paper presents a multimodal text classification method which fuses features extracted from the text and available metadata. Compared to the literature on multimodal classification methods, the novelty of the proposed method can be considered marginal due to the specific pipeline considered in the study. Furthermore, the experimental results on custom subsets of public datasets, which makes the study unreproducible, show that the proposed method without weighted averaging does not produce a significant performance boost compared to using transfer learning BERT with concatenated metadata and a random forest classifier. Thus, the performance improvement of the proposed method is mainly due to weighted averaging, a commonly used methodology for aggregating multimodal representations. Hence, the paper reaffirms that additional information in the form of metadata may boost the performance of multimodal text classification, which is known in the literature.